# Effective CRISPRa-mediated control of gene expression in bacteria must overcome strict target site requirements

Jason Fontana [1,5], Chen Dong [2,5], Cholpisit Kiattisewee [1], Venkata P. Chavali [3], Benjamin I. Tickman [1], James M. Carothers [1,3,4 ✉] & Jesse G. Zalatan [2,3,4 ✉]

In bacterial systems, CRISPR-Cas transcriptional activation (CRISPRa) has the potential to dramatically expand our ability to regulate gene expression, but we lack predictive rules for designing effective gRNA target sites. Here, we identify multiple features of bacterial promoters that impose stringent requirements on CRISPRa target sites. Notably, we observe narrow, 2–4 base windows of effective sites with a periodicity corresponding to one helical turn of DNA, spanning ~40 bases and centered ~80 bases upstream of the TSS. However, we also identify two features suggesting the potential for broad scope: CRISPRa is effective at a broad range of $\sigma^{70}$-family promoters, and an expanded PAM dCas9 allows the activation of promoters that cannot be activated by *S. pyogenes* dCas9. These results provide a roadmap for future engineering efforts to further expand and generalize the scope of bacterial CRISPRa.

[1] Molecular Engineering & Sciences Institute, University of Washington, Seattle 98195 WA, USA. [2] Department of Chemistry, University of Washington, Seattle 98195 WA, USA. [3] Department of Chemical Engineering, University of Washington, Seattle 98195 WA, USA. [4] Center for Synthetic Biology, University of Washington, Seattle 98195 WA, USA. [5] These authors contributed equally: Jason Fontana, Chen Dong. ✉email: jcaroth@uw.edu; zalatan@uw.edu

Developing tools to activate the expression of arbitrary genes has been transformative for biotechnology and biological research[1]. In metabolic engineering, regulating the timing and levels of the expression of complex multi-gene pathways is critical for reducing cellular burden and improving production of valuable metabolites[2]. To enable these goals, we recently developed a CRISPR-Cas transcriptional activation (CRISPRa) system that is effective in *Escherichia coli*. Our system can be combined with CRISPRi gene repression to programmably target multiple genes for simultaneous activation and repression[3]. Although our CRISPRa system can be used with heterologous genes, an outstanding challenge is to understand the rules that define effective target sites at arbitrary promoters in the genome.

To programmably downregulate target genes, we use nuclease defective Cas9 (dCas9) with a guide RNA (gRNA) that specifies a target site on the DNA. Targeting this complex to a promoter or an open reading frame (ORF) results in gene repression (CRISPRi)[4]. To enable simultaneous activation, we use modified guide RNAs, termed scaffold RNAs (scRNAs), that include a 3' MS2 hairpin to recruit a transcriptional activator fused to the MS2 coat protein (MCP)[3]. We can express multiple gRNAs and scRNAs to inhibit and activate genes simultaneously; gRNAs targeted to a promoter or ORF result in CRISPRi and scRNAs targeted to an appropriate site upstream of a minimal promoter result in CRISPRa.

We demonstrate here that the rules for targeting CRISPRa to effective sites in *E. coli* are surprisingly stringent. In prior work, we found that CRISPRa in *E. coli* was effective at target sites located in a narrow 40 base window between 60 and 100 bases upstream of the transcriptional start site (TSS)[3]. Here, we show that multiple factors combine to make the requirements for effective sites even stricter. We demonstrate that the basal promoter strength of the target gene and the sequence composition between the target site and the minimal promoter can have marked effects on gene activation. Further, by scanning the 40 base window at single base resolution, we find sharp peaks of activity and broad regions of inactivity that occur in a periodic 10–11 base pattern, corresponding to one helical turn along the DNA target. The observation that only a few precisely positioned target sites upstream of the TSS are effective for CRISPRa poses a significant challenge, as many genes will likely lack an NGG PAM sequence at exactly the right position necessary for *Streptococcus pyogenes* dCas9. These stringent requirements may explain why CRISPRa and other tools for gene activation in bacteria have lagged far behind comparable tools in eukaryotic systems, where such strict target site requirements are absent[5].

Although the requirements for bacterial CRISPRa target sites pose challenges, our data also demonstrate CRISPRa has the potential to be effective at a broad range of target genes. In addition to $\sigma^{70}$-dependent genes, CRISPRa can activate expression from genes that use the $\sigma^{70}$ family members $\sigma^{38}$, $\sigma^{32}$, and $\sigma^{24}$. We further demonstrate that the strict requirement for a precisely positioned PAM site can be partially overcome using a re-engineered dCas9 protein that targets an expanded set of PAM sequences[6]. Recently, some of the rules that we describe here were independently reported for an alternative bacterial CRISPRa system that can target genes regulated by $\sigma^{54}$ promoters[7]. Our results demonstrate that this behavior applies to a much broader range of $\sigma^{70}$ family promoters, which cover the majority of the *E. coli* genome[8]. The availability of these complementary systems should further extend the scope of bacterial CRISPRa. More broadly, by systematically defining the rules for effective CRISPRa sites, we identify strategies for improving and generalizing synthetic gene regulation in bacteria.

## Results

**A SoxS mutant reduces off-target activation.** Ideally, a synthetic transcriptional activator should only activate its programmed

target genes. The activation domain for our CRISPRa system is SoxS, a native *E. coli* transcription factor that directly binds DNA and activates endogenous gene targets as part of a stress response program[3]. We previously demonstrated that point mutations in the SoxS DNA-binding site can reduce activation of endogenous SoxS targets while maintaining CRISPRa activity at a heterologous reporter gene. However, the most effective single point mutants, R93A and S101A, did not completely abolish activity at endogenous targets. To further minimize off-target SoxS activity, we tested a double mutant SoxS(R93A/S101A). This double mutant SoxS retained full CRISPRa activity and showed a reduction in endogenous SoxS-dependent gene expression to levels indistinguishable from background (Fig. 1). Thus, SoxS (R93A/S101A) is an effective modular transcriptional effector that can activate gene expression only when recruited to a target gene via the CRISPR–Cas complex.

**A distance metric for target sites is not effective.** To determine whether we could predictably activate endogenous genes with CRISPRa, we selected three candidate genes with appropriately positioned PAM sites upstream of the TSS. Previously, we demonstrated that CRISPRa can activate heterologous promoters up to 50-fold with target sites positioned within a 40 base window between 60 and 100 bases upstream of the TSS[3]. We therefore targeted the CRISPR–Cas complex to the same window upstream of the candidate target genes. First, we targeted the *aroK-aroB* operon, which expresses enzymes involved in aromatic amino-acid biosynthesis, whose programmed overexpression could be useful for bioproduction[9]. Targeting the CRISPR–Cas complex to two sites within the optimal 40 base window resulted in no statistically significant increases in gene expression. Further, sites inside and outside of the 40 base window gave similar effects

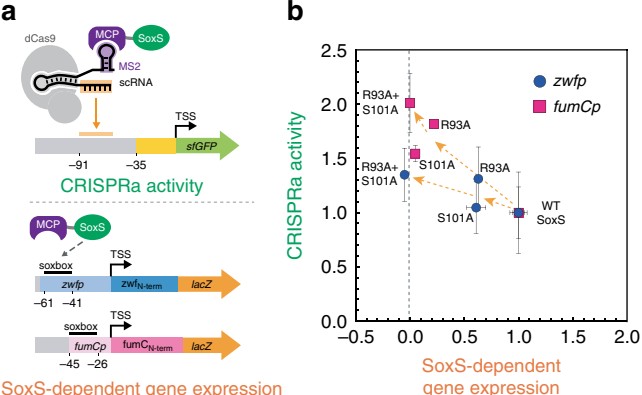

**Fig. 1 A SoxS double mutant maintains CRISPRa activity and does not activate endogenous SoxS targets. a** Reporter system for measuring the CRISPRa activity and endogenous SoxS-dependent gene expression of wild-type or mutant SoxS constructs. CRISPRa activity was determined in a strain harboring a genomically integrated sfGFP reporter (CD06, Supplementary Table 1). The endogenous SoxS-dependent gene expression was determined by monitoring *lacZ* expression from reporter plasmids where *lacZ* was driven by SoxS-regulated promoters *zwfp* and *fumCp*[44]. GFP fluorescence was measured by flow cytometry and *lacZ* activity was measured using a β-galactosidase assay. **b** SoxS(R93A/S101A) maintains CRISPRa activity and does not activate expression from the endogenous expression from the *zwfp* and *fumCp* reporters. Fluorescence and *lacZ* activity values were baseline-subtracted using a strain that does not express a scRNA. Both GFP levels and *lacZ* activities were normalized to the values observed in the strain with wild-type SoxS. Values represent the average ± standard deviation calculated from *n* = 3 biologically independent samples. Source data of **b** are provided as a Source Data file.

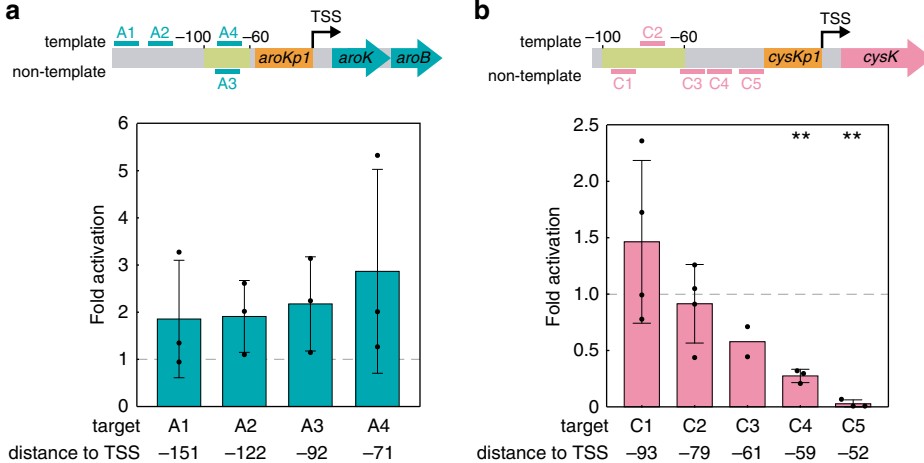

**Fig. 2 A simple distance metric does not predict CRISPRa activity. a** CRISPRa on the *aroK-aroB* operon. Two scRNA target sites within the 40 base window where CRISPRa is effective (−100 to −60) in heterologous reporter genes (A3–A4) and two sites further upstream (A1–A2) were chosen for the *aroKp1* promoter. **b** CRISPRa on the *cysK* gene. Three scRNA target sites within the 40 base window where CRISPRa is effective in heterologous reporter genes (C1–C3) and two sites further downstream (C4–C5) were chosen for the *cysKp2* promoter. The C4 and C5 sites resulted in repression; targeting these sites close to the core promoter may interfere with RNA polymerase binding. Gene expression was measured using RT-qPCR. Fold activation represents expression levels relative to a strain expressing an off-target scRNA (hAAVS1). In **a** and **b**, bars represent the average ± standard deviation calculated from $n = 4$ (C1, C2), $n = 3$ (A1–4, C4,C5), or $n = 2$ (C3) biologically independent samples. Some individual replicates in samples A1–4 and C1–2 appear to show activation, but a two-tailed unpaired Welch's *t* test indicates that the average differences relative to the off-target control are not statistically significant (*p* value > 0.05). Targeting CRISPRa to C4–5 resulted in repression, likely owing the short distance between the sites and the promoter. Stars indicate a statistically significant difference from the off-target control using a two-tailed unpaired Welch's *t* test (**\*\****p* value < 0.01). Exact *p* values: A1: 0.18, A2: 0.27, A3: 0.36, A4: 0.17, C1: 0.29, C2: 0.66, C3: 0.098, C4: 0.0022, C5: 0.00043. Source data are provided as a Source Data file.

(Fig. 2a). Next, we targeted *cysK*, an enzyme involved in cysteine biosynthesis[10]. Similar to what we observed with *aroK-aroB*, targeting three sites within the 40 base window resulted in no statistically significant increases in gene expression (Fig. 2b). Finally, we targeted *ldhA*, an enzyme involved in mixed acid fermentation[11]. We selected eight sites and observed no apparent relationship between the position of the target site and *ldhA* expression (Supplementary Fig. 1). Together, these results suggest that endogenous genes cannot be activated simply by targeting the CRISPR–Cas complex to sites positioned between 60 and 100 bases upstream of the TSS.

There are several possible explanations for our inability to activate endogenous bacterial genes with CRISPRa. First, we originally demonstrated CRISPRa using a relatively weak synthetic promoter. The basal levels of expression of endogenous genes vary significantly[12], and it may be difficult to increase the transcription of genes that are already strongly expressed[13]. In addition, some endogenous target genes might require an alternative sigma factor. Our original reporter gene is controlled by the $\sigma^{70}$ housekeeping sigma factor, and we do not know if our CRISPRa system is effective at gene targets that use alternative sigma factors. Another possibility is that native transcriptional regulator binding sites near endogenous gene promoters could disrupt CRISPRa. Finally, the optimal distance window metric that we previously identified may have been oversimplified. We initially identified the optimal window from an experiment with target sites spaced 10 bases apart, which may not be sufficient to generalize to any site within the 40 base window. To systematically explore these possibilities, we proceeded to test the efficacy of CRISPRa with a new set of synthetic promoters engineered with variable basal expression levels, alternative sigma factors, variable regulator binding sites, and variable scRNA target site positions.

**CRISPRa is sensitive to promoter strength**. To evaluate whether the intrinsic strength of the promoter affects CRISPRa, we tested

activation on a set of fluorescent reporter genes with minimal promoters spanning a 200-fold range in basal expression level (http://parts.igem.org) (Fig. 3a). We observed the most effective gene activation with a moderately weak J23117 promoter. With the weakest promoters, we could not detect any activation, even though their basal expression levels were only twofold weaker than the J23117 promoter. With stronger promoters, we observed progressively smaller CRISPRa-mediated activation of gene expression; the basal expression level increased, whereas the maximal, CRISPRa-induced expression remained roughly constant. These results indicate that the bacterial CRISPRa activity varies considerably with promoter strength, similar to effects observed in eukaryotic systems[14,15]. Thus, when targeting arbitrary endogenous genes, the level of activation that can be achieved may depend on the basal level of expression of its promoter.

**CRISPRa is effective with alternative sigma factors**. Bacterial transcription is initiated by a sigma factor binding to the minimal promoter and the RNA polymerase holoenzyme[16]. The SoxS activator binds directly to the α subunit of RNA polymerase[17], which suggests that our CRISPRa system could be compatible with genes that are controlled by non-housekeeping sigma factors. To investigate this possibility, we built synthetic promoters regulated by $\sigma^{38}$ (RpoS), $\sigma^{32}$ (RpoH), $\sigma^{24}$ (RpoE), and $\sigma^{54}$ (RpoN) to compare with our original housekeeping $\sigma^{70}$ (RpoD) promoter (Fig. 3b)[18–21]. CRISPRa was able to activate reporter gene expression when we targeted $\sigma^{38}$, $\sigma^{32}$, and $\sigma^{24}$-dependent promoters; these σ factors are all members of the $\sigma^{70}$ family. CRISPRa was not active on the $\sigma^{54}$ promoter, possibly because $\sigma^{54}$ initiates gene expression using a distinct mechanism that requires additional *cis*-regulatory elements[16]. These results suggest that CRISPRa can activate promoters regulated by non-housekeeping sigma factors such as $\sigma^{38}$, $\sigma^{32}$, and $\sigma^{24}$, and likely other members of the homologous $\sigma^{70}$ family.

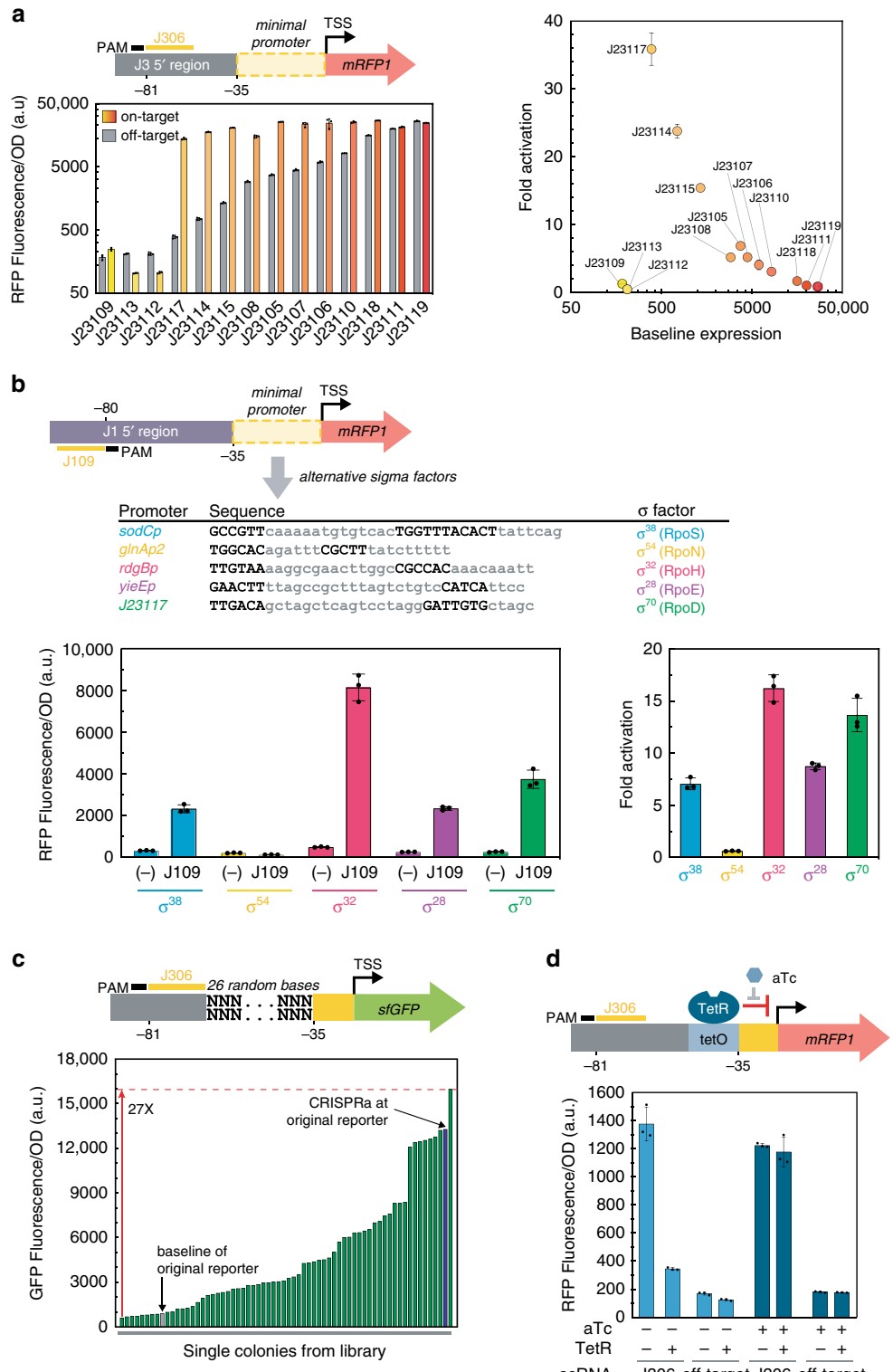

A recent paper described an alternative CRISPRa system that is capable of activating $\sigma^{54}$-dependent genes[7], which comprise a small fraction of the genome[8] (Supplementary Fig. 2). The availability of multiple, complementary CRISPRa systems should further extend the scope of bacterial CRISPRa. Both systems effectively activate expression from synthetic and heterologous promoters, and each system has the potential to target a different, non-overlapping set of endogenous genes.

**CRISPRa is sensitive to intervening sequence composition**. To determine whether the sequence composition between the target site and the −35 site affects CRISPRa, we constructed a promoter library with randomized sequences in this intervening region. We analyzed single colonies from this library and observed gene activation with a broad distribution over a 27-fold range (Fig. 3c). Although most variant sequences can still be activated (more than twofold) with CRISPRa, the large variation in activity was

**Fig. 3 CRISPRa is sensitive to promoter identity and local sequence. a** CRISPRa is sensitive to promoter strength. Promoters contain a scRNA target site at −81 from the TSS of the indicated J231NN minimal promoter, on the non-template strand[3]. The panel on the left shows the Fluorescence/OD$_{600}$ of strains expressing an on-target or off-target scRNA. The panel on the right shows the fold activation measured at each promoter relative to their baseline expression with an off-target scRNA (J206). **b** CRISPRa can activate promoters regulated by σ$^{38}$ (RpoS), σ$^{32}$ (RpoH), and σ$^{24}$ (RpoE) sigma factors. The minimal promoter from the reporter plasmid was replaced with *sodCp*, *glnAp2*, *rdgBp*, or *yieEp*. The −35 and −10 regions are highlighted in bold. The plot on the left shows the Fluorescence/OD$_{600}$ when CRISPRa targeted each promoter at the J109 target site (−80 from the TSS on the template strand) or with an off-target scRNA (hAAVS1, labeled (−)). The plot on the right shows the fold activation measured at each promoter relative to an off-target scRNA (J206). **c** CRISPRa activity differs significantly among promoters with varying sequence composition between the scRNA target and the −35 region. Green bars represent the Fluorescence/OD$_{600}$ of overnight cultures from individual colonies. The blue bar represents the Fluorescence/OD$_{600}$ of a strain expressing the J3-J23117-sfGFP reporter, activated by CRISPRa with the J306 scRNA. The gray bar represents a negative control expressing the J3-J23117-sfGFP reporter plasmid with CRISPRa targeting an off-target site (J206). **d** CRISPRa was inhibited binding of the TetR transcriptional repressor binding to a tet operator (tetO) site placed upstream of the −35 region. Cultures where CRISPRa was targeted to the J306 site or to an off-target site (J206) were grown overnight in media ±1 μM aTc. In panels **a**, **b**, and **d**, values represent the average±standard deviation calculated from *n* = 3 biologically independent samples. **c** Bars represent the value of *n* = 1 biologically independent samples. Source data are provided as a Source Data file.

unexpected because each reporter gene was driven by the same minimal promoter and contained the same scRNA target site. One possible interpretation of this result is that these randomized intervening sequences contain binding sites for endogenous transcriptional regulators; there is evidence that binding sites can emerge with relatively high frequency from random sequences[22]. These sites could potentially affect CRISPRa by directly blocking access to a scRNA target site, by blocking RNA polymerase binding, or by interfering with the ability of a CRISPRa effector protein to engage with RNA polymerase.

To directly test the hypothesis that a bound transcriptional effector can disrupt CRISPRa, we introduced a binding site for the transcriptional repressor TetR upstream of the −35 region[23]. The presence of a bound TetR significantly disrupted CRISPRa-mediated gene activation. Further, adding anhydrotetracycline (aTc), which releases TetR from the DNA, restored CRISPRa activity to the levels observed when TetR was not present (Fig. 3d). Because endogenous genes contain binding sites for a variety of transcriptional activators and repressors upstream of the minimal promoter[24,25], this effect could be contributing to the inconsistent and variable effects we observed when targeting endogenous genes for CRISPRa (Fig. 2).

To determine whether transcription factor-binding sites appear in the library of randomized intervening sequences, we sequenced 29 variants spanning the full range of observed activation levels (Supplementary Table 6). Only five intervening sequences contained exact matches to a known consensus transcription factor-binding motif. However, all sequences contained at least one match within a single base of a known motif, and it is well established that DNA-binding proteins can recognize sites that deviate from the consesus[26]. There was no significant correlation between gene activation by CRISPRa and the number of these motifs (Spearman rank order correlation $r_s$ = 0.29, $p$ = 0.11, Supplementary Fig. 3A), but we note that it is not known which of these motifs actually bind endogenous transcription factors. We did find that intervening sequences that give more effective CRISPRa tend to be more GC-rich, though we do not yet understand the basis for this trend ($r_s$ = 0.42, $p$ = 0.02, Supplementary Fig. 3B & C). Nonetheless, these experiments indicate that the composition of the intervening sequence between the CRISPR–Cas complex and the minimal promoter is an important factor determining the level of CRISPRa.

**CRISPRa is sharply dependent on single base shifts**. Our original hypothesis that optimal target sites are located −60 to −100 bases upstream of the TSS was based on an experiment with scRNA sites spaced every 10 bases[3]. To further test this hypothesis, we targeted the CRISPRa complex to a window from −61 to −113 at single base resolution. We used a reporter gene with five scRNA sites located at −61, −71, −81, −91, and −101 relative to the TSS, and we inserted 1–12 bases upstream of the −35 site to generate a set of reporter genes that allowed the CRISPRa complex to target every possible distance in the optimal targeting window. Using this reporter gene set, we found that shifting the target site by 1–3 bases caused significant decreases in activation (Fig. 4a). Shifting the target site further by 4–9 bases decreased expression to levels nearly indistinguishable from background. At 10–11 base shifts, corresponding to one full turn of a DNA helix, gene expression increased again. This periodic positional dependence of CRISPRa extended over the entire −60 to −100 window, with the strongest peaks centered at −81 and −91 and smaller peaks centered at −102 and −70. There is no recovery of activity when the site at −101 is shifted to −111, outside of the −60 to −100 window. This sharp periodic relationship suggests that the criteria for effective target sites are quite stringent, and that both distance and relative periodicity to the TSS are critical factors.

Notably, the distance to the TSS is not the sole determining factor for CRISPRa-mediated expression level. Sites that overlap at the same distance, such as the original −81 site and the −71 site shifted by 10, do not give the same gene expression output (Fig. 4a). These discrepancies could arise from intrinsic differences in the activity of the 20 base scRNA target sequence (Supplementary Fig. 4) or from the effect of different intervening sequence composition between the scRNA target site and the minimal promoter (Fig. 3).

Because we demonstrated that sequence composition can have unexpected effects on CRISPRa (Fig. 3), we tested whether the periodicity of CRISPRa was similar in different sequence contexts. We obtained comparable periodic phase dependence when different nucleotide sequences were used to shift the scRNA target site, and when the bases were inserted at a different location in the promoter (Supplementary Fig. 5A). Similar results were also obtained when we performed the base shift experiment with a reporter that had a different 5' upstream sequence (Supplementary Fig. 5B) or where the minimal BBa_J23117 promoter was replaced by endogenous *aroK* promoter (Supplementary Fig. 5C). Further, the sharp positioning dependence was observed when targeting the template or non-template strand of the reporter (Supplementary Fig. 5D). Finally, one possible confounding effect could arise if the basal expression level of the reporter gene changes when bases are inserted, which can affect the efficacy of CRISPRa (Fig. 3a). However, we observed that basal expression from the original reporter and the +5 base shifted reporter were indistinguishable (Supplementary Fig. 5E). Together, these experiments confirm that bacterial CRISPRa is sensitive to periodicity in multiple different sequence contexts.

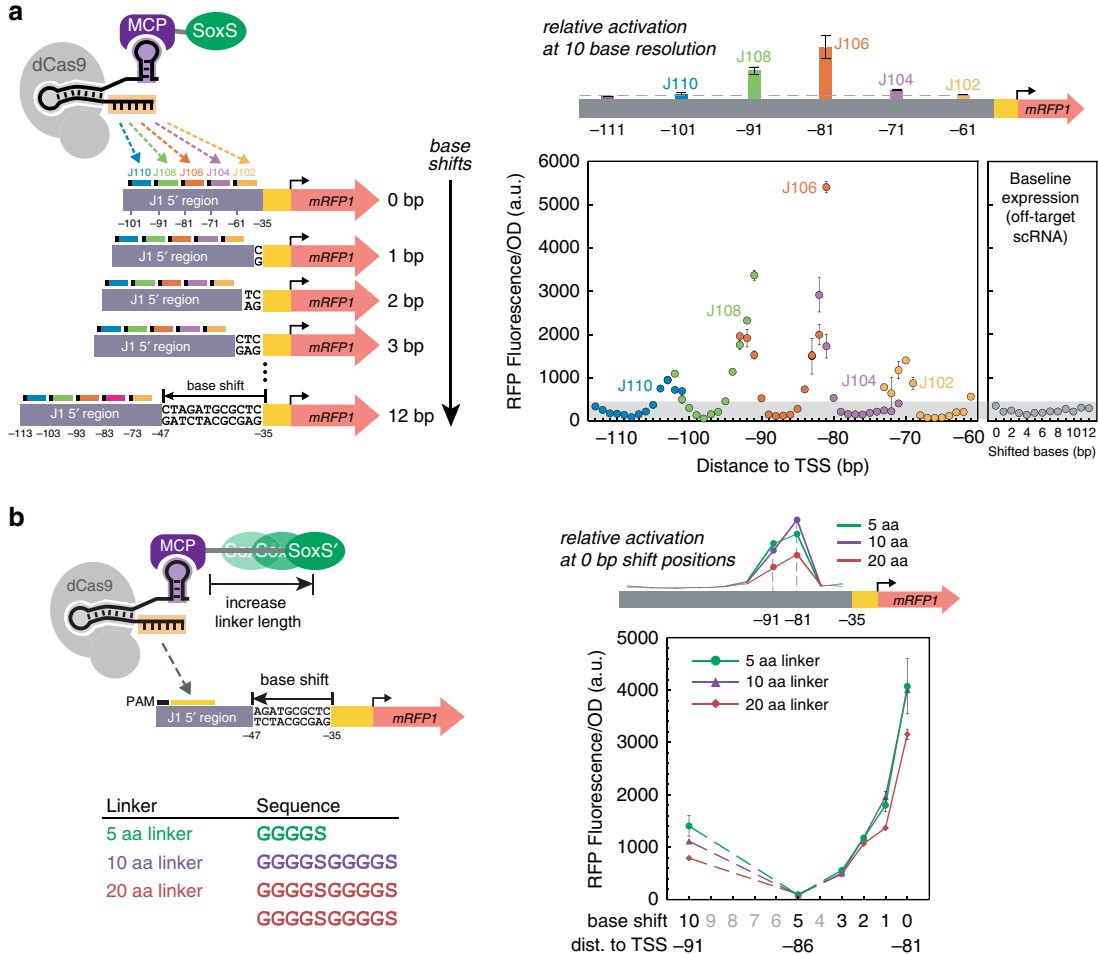

**Fig. 4 CRISPRa is sensitive to the precise position of the scRNA target. a** CRISPRa displays periodic positioning dependence with peak activities every 10–11 bases between −60 and −100 from the TSS. Reporter genes were constructed by inserting 0–12 bases upstream of the −35 region of the J1-J23117-mRFP1 reporter. Five scRNA sites (J102, J104, J106, J108, J110) with positions −61, −71, −81, −91, −101 from the TSS on the non-template strand of the original promoter were targeted. In this way, the complete −61 to −113 region can be covered at single base resolution. The color coding indicates data for the same target site shifted across a 12 base window. The panel on the right shows the baseline expression of reporters with shifted bases when an off-target scRNA was used (J206). The gray area represents the range of the baselines among the reporter series. For comparison, previous CRISPRa data for the J102, J104, J106, J108, J110 target positions at 10 base resolution are shown on the schematic above the plot[3]. **b** Extending the linker length between MCP and SoxS does not change the position dependence of CRISPRa. The J1-J23117-mRFP1 reporter plasmid series with base shifts were delivered together with CRISPRa components for targeting J106. The MCP-SoxS(R93A) effector contained 5aa, 10aa, and 20aa linkers. For comparison, previous CRISPRa data with 5aa, 10aa, or 20aa linker between MCP and SoxS targeting at the −81 and −91 positions are shown on the schematic above the plot[3]. Values in **a** and **b** represent the average±standard deviation calculated from n = 3 biologically independent samples. Source data are provided as a Source Data file.

In the experiments described above, comparisons between single base shifted scRNA sites were performed with different reporter gene constructs, each with a differing number of inserted bases. To test the positional dependence of CRISPRa at single base resolution in a single reporter construct, we designed an alternative reporter gene with 6 adjacent scRNA target sites between −81 and −86. We again observed sharp drops in gene expression when targeting sites one or more bases away from the optimal site at −81 (Supplementary Fig. 5F).

The finding that CRISPRa displays the same ~10 base periodicity as the DNA helix suggests that the angular phase of the CRISPRa complex relative to the minimal promoter is critical for effective activation. Our bacterial CRISPRa system requires a direct interaction between the SoxS activation domain and RNA polymerase[3], and this interaction appears to be highly sensitive to both the distance and relative phase of the target site to the minimal promoter. The sharp phase dependence of CRISPRa may be a general feature of transcriptional regulation

in *E. coli*. The native SoxS protein and other transcription factors such as CAP and LacI have restrictive positioning requirements that correspond to DNA periodicity[27–34]; we confirmed this result with an endogenous SoxS reporter (Supplementary Fig. 6). In practice, this periodic behavior means that effective target sites must be located at one of the narrow peaks of activation within the optimal distance range. These stringent requirements suggest that targeting endogenous genes will be extremely challenging. There is ~1 PAM site every 10 bases in the regions upstream of endogenous promoters in *E. coli* (Supplementary Fig. 7A & B), and the likelihood that a PAM site will be located at the appropriate phase within a 10 base window is low (Supplementary Fig. 7C).

**Tuning structure to expand target site range is ineffective.** If rotating the CRISPRa complex out of phase along the DNA prevents SoxS from interacting with RNA polymerase, then a

longer amino-acid linker to SoxS might allow effective CRISPRa at more scRNA sites. To test this possibility, we extended the linker between MCP and SoxS from five amino acids (aa) to 10 or 20 aa, but even with these longer linkers we observed the same sharp dependence on the target site position as with the original 5 aa linker (Fig. 4b). We obtained similar results using a linker with a different amino-acid composition (Supplementary Fig. 8A).

Another potential approach to expand the range of effective CRISPRa sites would be to change the spatial position of the MCP-SoxS protein by altering the position of the MS2 hairpin that binds MCP. We therefore tested multiple alternative scRNA designs that present the MS2 hairpin at different locations. Extending the MS2 stem by 2, 5, 10, and 20 bp resulted in progressively lower CRISPRa activity, but no change in the position of the target sites that were most effective (Supplementary Fig. 8B). Similarly, no changes were observed with alternative scRNA designs with one or two MS2 hairpins presented from different locations within the scRNA structure (Supplementary Fig. 8C).

Finally, we assessed whether any alternative activation domains could produce a different phase dependent behavior. Previously, these constructs all produced weaker activation than SoxS[3], perhaps because they have each distinct optimal target site positions. We tested MCP fused to TetD, αNTD, lambda cII, and RpoZ[3], and dCas9 fused to RpoZ[35]; however, none of these constructs produced gene activation at any site that was not already effective with SoxS (Supplementary Fig. 9).

Although endogenous bacterial transcription factors exhibit a sharp periodic dependence on distance[27–34], it remains surprising that no structural modifications of the CRISPRa complex produced any changes in the phase dependence. If SoxS is simply tethered to the CRISPRa complex by a flexible linker, we would have expected the peak of effective CRISPRa sites to broaden with longer linkers. The failure of this prediction suggests that our understanding of the CRISPR–Cas complex and its interactions with bacterial transcriptional machinery is fundamentally incomplete, or that the linker tethering SoxS to the CRISPRa complex is not truly flexible. Practically, it means that we still lack a way to expand the range of effective CRISPRa target sites.

**A dCas9 variant expands the range of targetable sites**. Because there is a limited number of genes with an appropriate NGG PAM site at precisely the optimal position upstream of the promoter (Supplementary Fig. 7C), we attempted to expand the scope of targetable PAM sites for CRISPRa. We used a recently characterized dCas9 variant, dxCas9(3.7), that has improved activity at a variety of non-NGG PAM sites including NGN, GAA, GAT, and CAA[6]. We generated reporter plasmids by replacing AGG PAM sites with alternative PAM sequences and delivered a CRISPRa system with dxCas9(3.7) to target these reporters. dxCas9(3.7) maintained the ability to target the AGG PAM and showed significantly increased levels of activation at alternative PAM sites compared to dCas9 (Fig. 5a). Activation levels varied with different PAM sites and correlated well with dxCas9(3.7) activity previously reported in human cells (Supplementary Fig. 10A)[6]. dxCas9(3.7) showed similar distance and phase dependent target site preferences as dCas9 (Supplementary Fig. 10B & C), but its expanded PAM scope makes it more likely that an arbitrary gene will have a targetable PAM site at an effective position. Bioinformatic analysis of the sequences between transcriptional units in E. coli revealed that there are on average 6.4 times more dxCas9(3.7)-compatible PAM sites than NGG PAM sites (Supplementary Fig. 10D). Accounting for the fact that dCas9 has some activity at non-NGG sites[6] (Fig. 5a), there are still on average ~2.2-fold more dxCas9(3.7)-compatible

PAM sites than dCas9-compatible PAM sites (Supplementary Fig. 10D).

To demonstrate the utility of dxCas9(3.7) for CRISPRa at sites inaccessible to dCas9, we constructed a reporter plasmid that contains an AGG PAM site at the original position with maximum CRISPRa activity and an AGT PAM five bases downstream. Using this reporter, we observe that both dCas9 and dxCas9(3.7) are effective for CRISPRa at the optimally positioned NGG PAM site, but neither is capable of activating the AGT PAM site, which is five bases out of phase from the optimal site (Fig. 5b). We then inserted five bases into the reporter to shift the AGT PAM site into the peak activation range. With this reporter, neither dCas9 nor dxCas9(3.7) can activate the NGG PAM site, which is now out of phase. dxCas9(3.7) was now able to effectively activate the AGT PAM site, and dCas9 was ineffective at this site (Fig. 5b). This result confirms that dxCas9 (3.7) is able to activate optimally positioned target sites that are inaccessible to dCas9. We expect that this behavior will be effective at many σ[70]-family promoters (Fig. 3b), and a recent report demonstrated a similar behavior of dxCas9(3.7) at σ[54]-dependent promoters[7].

**Defined rules enable endogenous gene activation**. Our systematic characterization of the requirements for effective CRISPRa in E. coli demonstrates that candidate genes must have a targetable PAM site located at one of the sharp peaks of activity upstream of the TSS. In hindsight, the scRNA sites at endogenous genes that we initially targeted in Fig. 2 did note meet this criterion. To determine whether the revised rules would enable activation of endogenous E. coli genes, we surveyed the genome for candidate genes with appropriately positioned, dxCas9(3.7)-compatible PAM sites (Supplementary Methods) (Supplementary Fig. 7C). We selected candidates with multiple potentially effective PAM sites and further narrowed the pool based on two additional criteria: (1) genes should not be too highly expressed (Fig. 3a) and (2) genes should be regulated by σ[70], which is the sigma factor that regulates most genes[8] (Fig. 3b). Ideally, we would also exclude genes with tightly bound transcriptional regulators in the promoter region (Fig. 3d), but this information is not readily available. We chose six genes that could be tested using reporter strains from the E. coli promoter collection[36] and targeted two PAM sites for each gene.

We first examined the yajG gene, which had two plausible target sites, one of which was only compatible with dxCas9(3.7). We also included an additional site predicted to be out of phase and ineffective for CRISPRa. We observed significant, ~4–6-fold gene activation for the two sites located at the predicted peak of activity at −80/−81, and no activation at the out of phase site at −87 (Fig. 6a). The site at −81 is inaccessible to dCas9, and we only observed activation with dxCas9(3.7). We proceeded to test an additional five genes with partial success. We observed significant activation at poxB (~10-fold) and uxuR (approximately twofold) (Fig. 6b). We validated these results by performing RT-qPCR on the endogenous yajG and poxB loci. Targeting CRISPRa to these genes resulted in increases in RNA levels (Supplementary Fig. 11). Targeting CRISPRa to araE produced a statistically significant difference in expression, but the activation measured was modest (1.13-fold). For the remaining two candidate genes, ansB was modestly repressed at one of the target sites and we did not observe a statistically significant difference in expression at ppiD. Similarly, one of the ldhA sites that we targeted in initial experiments (Supplementary Fig. 1) was at a predicted optimal site at −91 and failed to give substantial activation. Thus, of seven endogenous genes tested with target sites that we predict should be effective

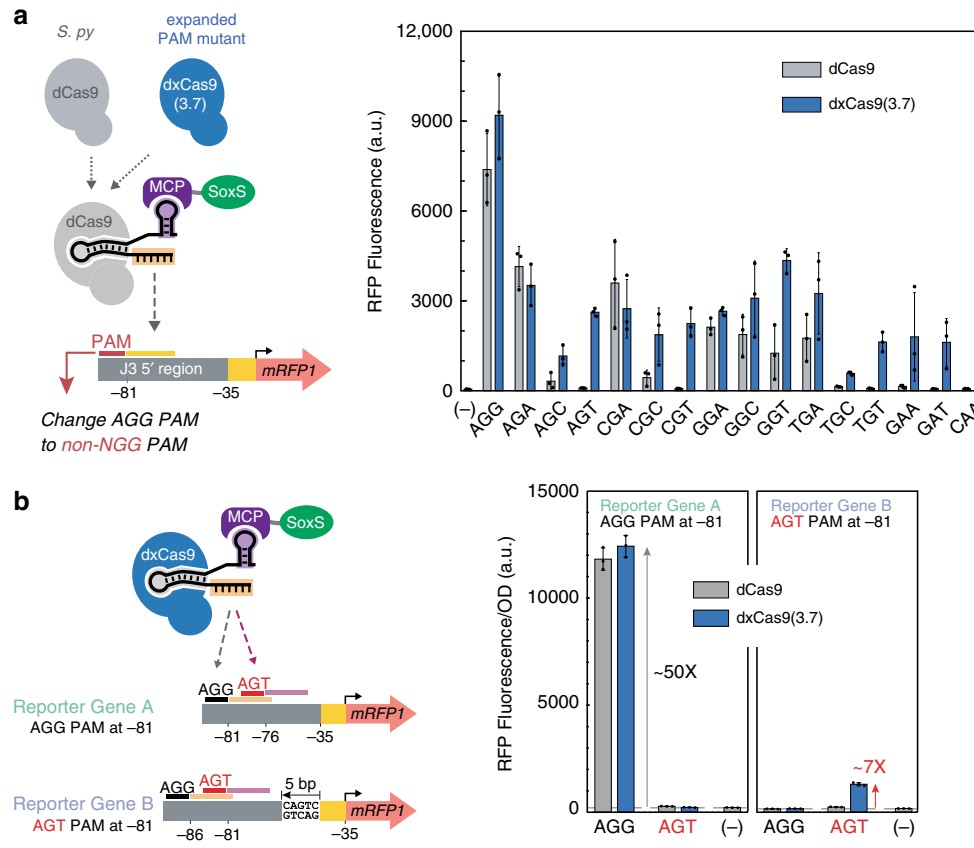

**Fig. 5 dxCas9(3.7) expands the range of targetable scRNA target sites by recognizing alternative PAMs. a** CRISPRa with dxCas9(3.7) displayed activity on non-NGG PAM sites with AGA, AGC, AGT, CGA, CGC, CGT, GGA, GGC, GGT, TGA, TGC, TGT, GAA, GAT, CAA sequences. CRISPRa activity with dxCas9(3.7) on non-NGG PAM sites was generally lower (6-fold to 89-fold activation relative to a control without a scRNA) compared to the AGG PAM site (188-fold activation). *Sp*-dCas9 also displayed moderate CRISPRa activity at non-NGG PAM sites with AGA, CGA, GGA, GGC, GGT, TGA sequences, consistent with published reports[6]. Reporter plasmids were constructed by replacing the AGG PAM site for the J306 target in the J3-J23117-mRFP1 reporter with alternative PAM sequences that have been previously reported to be recognized by dxCas9(3.7) in human cells[6]. The (−) sign indicates a control expressing the original reporter with the AGG PAM and the CRISPRa components with *Sp*-dCas9, the activation domain and no scRNA. **b** dxCas9 (3.7) can activate promoters that cannot be activated by *Sp*-dCas9. When the scRNA target at the optimal position (−81 to the TSS) has an AGG PAM site, both *Sp*-dCas9 and dxCas9(3.7) increased gene expression by 50-fold. When the scRNA target at the optimal position has an AGT PAM site, only dxCas9 (3.7) displayed a sevenfold increase in gene expression while *Sp*-dCas9 was inactive. The reporter gene has a target with an AGG PAM (M1) and a target with an AGT PAM (M2) upstream of a BBa_J23117 minimal promoter. In reporter gene A, the AGG target was located −81 to the TSS on the non-template strand and the AGT target was located −76 to the TSS on the non-template strand. In reporter gene B, 5 bases were inserted upstream of the −35 region, shifting the locations of the AGG target and AGT target to −86 and −81, respectively. The (−) sign indicates a negative control strain that contains the reporter plasmid and a plasmid expressing Sp-dCas9, the activation domain and an off-target scRNA (J206). Bars in **a** and **b** represent the averagerom the *E. coli* promoter±standard deviation calculated from *n* = 3 biologically independent samples. Source data are provided as a Source Data file.

(the six genes from Fig. 6b and *ldhA* from Supplementary Fig. 1), we were able to activate three genes with more than twofold increases in gene expression.

Although any success at endogenous gene activation is encouraging, significant challenges remain for predictable CRISPRa in bacteria. Our results suggest that even with a precise distance metric for effective target sites, some genes will not be predictably activated. There are several possible explanations: (1) tightly bound negative regulators could interfere with CRISPRa (Fig. 3d), and (2) small errors in TSS annotation could lead to inaccurate predictions for effective sites, given that 1–2 base shifts can have dramatic effects on CRISPRa (Fig. 4), and (3) intrinsic differences in the activity of the 20 base scRNA target sequence (Supplementary Fig. 4).

## Discussion

Bacterial CRISPRa is sensitive to a number of factors, including (i) the strength of the target promoter, (ii) the sigma factor regulating the promoter, (iii) the sequence composition immediately upstream of the minimal promoter, (iv) the composition of the scRNA target sequence, (v) the position of the scRNA target site with respect to the TSS at single base resolution. Some of these factors, such as promoter strength and scRNA target sequence composition, are also relevant in eukaryotic systems[13,15,37,38]. Other factors are plausible given our understanding of bacterial transcription. Sigma factor levels are regulated to control gene expression in response to cell state and external signals[16], so it is reasonable that we observed variable levels of activation from promoters with alternative sigma factors. Many bacterial genes are controlled by negative regulators[39], and different sequences upstream of the minimal promoter could be recruiting repressors.

The most unexpected property that we observed with bacterial CRISPRa was its sharp, periodic dependence on-target site position. This behavior is quite distinct from CRISPRa in eukaryotes, where a broad range of sites upstream of the TSS are effective[40], possibly because eukaryotic activators typically recruit transcription factors

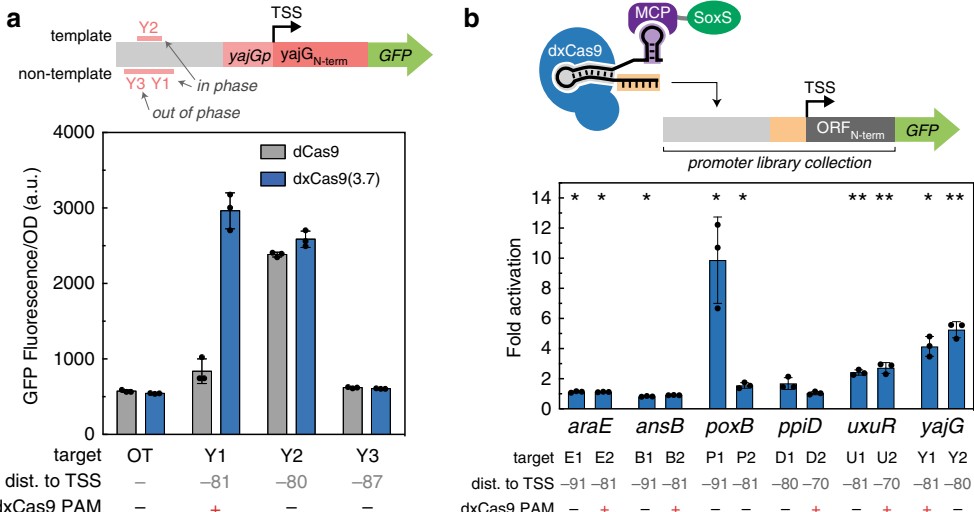

**Fig. 6 Predictive rules enable endogenous activation. a** CRISPRa using dCas9 and dxCas9(3.7) was targeted to a *yajG* reporter plasmid from the *E. coli* promoter collection[36]. Three scRNA target sites were selected; two sites were located at the positions where CRISPRa was most effective (Y1–2), and one was located out of phase (Y3). A negative control (OT) expressing an off-target scRNA (J306) was included. **b** CRISPRa was targeted to *yajG* and five additional promoters from the *E. coli* promoter collection (Supplementary Methods). Two scRNA sites located at the positions where CRISPRa was most effective were targeted for each gene using dxCas9(3.7). Samples are arranged by baseline expression of the target genes, in ascending order left to right. Fold activation indicates the median fluorescence of strains relative to an off-target control (J306). Values **a** and **b** represent the average±standard deviation calculated from $n = 3$ biologically independent samples. Stars indicate a statistically significant difference from the off-target control using a two-tailed unpaired Welch's *t* test (\**p*-value < 0.05, \*\**p* value < 0.01). Exact *p* values: E1: 0.036, E2: 0.024, B1: 0.031, B2: 0.141, P1: 0.033, P2: 0.021, D1: 0.088, D2: 0.585, U1: 0.0008, U2: 0.001, Y1: 0.013, Y2: 0.003. Source data are provided as a Source Data file.

and chromatin modifying machinery rather than directly recruiting RNA polymerase. There is precedent for bacterial transcriptional activators that are sensitive to target site periodicity[27–34], but the marked changes in activity with only single base shifts is surprising. Moreover, it is puzzling that we were unable to predictably alter or broaden the range of sites that are effective. Our models for how activators interact with bacterial transcription machinery may be incomplete. It will likely be productive to continue screening for activity at out-of-phase target sites using additional systematic modifications to the CRISPRa complex structure, alternative CRISPR–Cas systems, and additional candidate transcriptional activation domains.

Despite the challenges described above for identifying effective CRISPRa sites in *E. coli*, our systematic characterization provides a framework for immediate practical applications and a path for future improvements. We now have a clear understanding of the criteria needed to design synthetic promoters that can be regulated by CRISPRa, which will enable the construction of complex, tunable synthetic multi-gene circuits. To extend the scope of CRISPRa to endogenous target genes, expanded PAM variants like dxCas9(3.7)[6], or orthologous dCas9 proteins with alternate PAM specificities[41,42] will open more DNA sites for targeting, increasing the likelihood of finding a targetable site at an optimal position relative to the TSS. These strategies lay the groundwork for more widespread use of bacterial CRISPRa in basic research and practical applications including functional genomics screens, metabolic engineering, and synthetic microbial communities.

## Methods
**Bacterial strain construction and manipulation**. Plasmids were cloned using standard molecular biology protocols. Bacterial strains with sfGFP or mRFP1 reporter strains are described in Supplementary Table 1. The CRISPRa system used for each figure panel is described in Supplementary Table 2. Guide RNA target sequences are described in Supplementary Table 3. Plasmid containing the reporter genes and the CRISPR components are described in Supplementary Table 4. *S. pyogenes* dCas9 (*Sp*-dCas9) or dxCas9(3.7) were expressed from the endogenous

*Sp.pCas9* promoter in a p15A vector. MCP-SoxS containing wild-type and mutant SoxS were expressed using the BBa_J23107 promoter (http://parts.igem.org) in the same plasmid with dCas9. The scRNAs were expressed using the BBa_J23119 promoter, either in the same plasmid with the dCas9 protein and the activation domain or in a separate ColE1 plasmid. The scRNA.b1 or scRNA.b2 designs, where the endogenous tracr terminator hairpin upstream of MS2 was removed[3], were used in all experiments except otherwise noted. The *zwfp-lacZ* and *fumCp-lacZ* reporter plasmids were generated in a previous study[3]. mRFP1 and sfGFP reporters were expressed from the weak BBa_J23117 minimal promoter (http://parts.igem.org) in a low-copy pSC101\*\* vector. Variant versions of reporter genes are described in the Supplementary Methods. Plasmid libraries containing N26 sequences between the scRNA target site and BBa_J23117 minimal promoter were constructed by PCR amplification using mixed bases oligos (IDT). The dxCas9(3.7)-VPR plasmid was a gift from David Liu (Addgene #108383)[6].

**Flow cytometry**. Single colonies from LB plates were inoculated in 500 μL EZ-RDM (Teknova) supplemented with appropriate antibiotics and grown in 96-deep-well plates at 37 °C and shaking. Cultures were grown overnight at 37 °C and shaking and then diluted in 1:50 in Dulbecco's phosphate-buffered saline and analyzed on a MACSQuant VYB flow cytometer with the MACSQuantify 2.8 software (Miltenyi Biotec). A side scatter threshold trigger (SSC-H) was applied to enrich for single cells until 10000 events were collected. The FlowJo 10.0.7 software was used to apply a narrow gate along the diagonal line on the SSC-H vs SSC-A plot was selected to exclude the events where multiple cells were grouped together. Within the selected population, events that appeared on the edges of the FSC-A vs. SSC-A plot and the fluorescence histogram were excluded.

**Plate reader experiments**. Single colonies from LB plates were inoculated in 500 μL EZ-RDM (Teknova) supplemented with appropriate antibiotics and grown in 96-deep-well plates at 37 °C and shaking overnight. For experiments with the *E. coli* promoter collection[36] the activation domain was placed under the control of a tet-inducible promoter. Attempts to use constitutive CRISPRa were unsuccessful due to plasmid instability, possibly because of toxicity arising from increased expression of the target genes. Single colonies from LB plates were inoculated in 500 μL EZ-RDM supplemented with appropriate antibiotics and 400 nM anhydrotetracycline (aTc) and grown in 96-deep-well plates at 37 °C and shaking overnight. 150 μL of the overnight culture were transferred into a flat, clear-bottomed black 96-well plate and the $OD_{600}$ and fluorescence were measured in a Biotek Synergy HTX plate reader and analyzed using the BioTek Gen5 2.07.17 software. For mRFP1 detection, the excitation wavelength was 540 nm and emission wavelength was 600 nm. For sfGFP detection, the excitation wavelength was 485 nm and emission wavelength was 528 nm.

**Quantitative RT-PCR**. Single colonies from LB plates were inoculated in 5 mL LB containing appropriate antibiotics and grown overnight at 37 °C and shaking. Overnight cultures were diluted 1:100 into 5 mL EZ-RDM supplemented with appropriate antibiotics and grown at 37 °C and shaking until an $OD_{600}$ of 0.5 (using 150 μL of culture in a 96-well plate) was reached. For the experiments targeting *yajG* and *poxB*, the activation domain was placed under the control of a tet-inducible promoter and cultures in EZ-RDM were supplemented with 400 nM aTc. Cultures were pelleted and total RNA was extracted using the Aurum Total RNA Mini Kit (Bio-rad). Reverse transcription reactions were performed from 1 μg RNA in 20 μL reactions using iScript reverse transcriptase (Bio-Rad). qPCR reactions were prepared in triplicate in a final volume of 10 μL using SsoAdvanced Universal SYBR Green Supermix (Bio-Rad), 0.5–5 ng of cDNA and 400 nM primers. The reaction was performed in a CFX Connect (Bio-Rad) with a 58 °C annealing temperature and 30 s extension time. A list of the qPCR primer sequences is provided in Supplementary Table 5. Expression levels for each gene were calculated by the Bio-Rad CFX Maestro 4.0.23225.0418 software by normalizing to the 16S rRNA gene and relative to a negative control carrying an off-target scRNA using the $\Delta\Delta CT$ method[43].

**Statistics and reproducibility**. Statistical significance was calculated using two-tailed unpaired Welch's *t* tests. To ensure reproducibility, experiments were performed using $n = 3$ biologically independent samples, unless otherwise noted.

**Reporting summary**. Further information on research design is available in the Nature Research Reporting Summary linked to this article.

## Data availability
Data supporting the findings of this work are available within the paper and its Supplementary Information files. A reporting summary for this article is available as a Supplementary Information file. The data sets generated and analyzed during the current study are available from the corresponding author upon request. The source data underlying Figs. 1b and 2–6, as well as Supplementary Figs. 1, 2, 3A–B, and 4–11 are provided as a Source Data file.

## Code availability
Custom Python code to generate the DNA sequences between transcriptional units in *E. coli* and analyze the density of PAM sites in these sequences (detailed in the Supplementary Methods) is available on GitHub (https://github.com/carothersresearch/Fontana-Dong_2020_NatComm).

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

## Acknowledgements

The dxCas9(3.7)-VPR plasmid was a gift from David Liu (Addgene #108383). We thank Mary Lidstrom, Joanne Wong, Semira Beraki, and members of the Zalatan and Carothers groups for technical assistance, advice, and helpful discussions. This work was supported by NSF Award 1817623 (J.M.C, J.G.Z.) and NSF Award 1844152 (J.M.C.).

## Author contributions

J.F., C.D., C.K., B.I.T., J.M.C., and J.G.Z. designed experiments and analyzed data. J.F., C.D., C.K., V.P.C., and B.I.T. performed experiments. J.F., C.D., J.M.C., and J.G.Z. wrote the manuscript.

## Competing interests

The authors declare no competing interests.
