## [Peer Review File · Nature Communications]

Reviewers' Comments:

Reviewer #1:

Remarks to the Author:

In their manuscript, "Effective CRISPRa-Mediated Control of Gene Expression in Bacteria Must Overcome Strict Target Site Requirements", Fontana and collaborators present new insights into the use of a dCas9 tethered SoxS effector as a bacterial CRISPRa system. Despite some interesting observations, their findings are fundamentally negative results that call into question the existence of any reasonable use cases for CRISPRa. Additionally, the authors' results about the extreme spacing dependence of CRISPRa should have been expected, and frankly should have been addressed in their initial (2018) paper. I detail these points further below.

1. The authors' further characterization of CRISPRa undermines many use cases for CRISPRa. The clearest use case for CRISPRa over another method of over-expression (such as cloning onto a plasmid, or cloning in a different promoter) is for screening.

a. The need to target promoters diminishes the value of CRISPRa. To be able to perform useful CRISPRa experiment, one must know the promoters of the genes that one wants to target. Even well studied organisms like *E. coli* have incompletely characterized promoters. Many genes have multiple (sometimes overlapping) promoters. Targeting the wrong promoter could readily lead to repression (by interfering with the active promoter) instead of activation.

b. The authors show that the sequence between -60 and -35 makes a huge difference in CRISPRa activity but provide no information (other than speculation) on how this could happen. So even if all of the above criteria could be met, one would still have no way of knowing whether the system was actually working.

c. Because of the strict target site requirements, only some genes have PAMs that could be successfully targeted. This is partially addressed by dxCas9. However, off-target effect of this variant in CRISPRa system should be evaluated.

2. It should have been apparent that strict spacing would be required for SoxS function. Almost all Class I activators (those that don't overlap the -35) function by interacting with the C-terminal domain of the alpha subunit of RNAP (αCTD). An extensive body of literature has shown that these activators must be on the same face of the DNA helix as RNAP to function. For example, in (doi:10.1093/nar/gku625), the ability of CRP to activate a promoter across different spacings is tested and shown to have DNA periodicity. Similarly, in (<https://doi.org/10.1046/j.1365-2958.1999.01599.x>), it is shown that activation by MarA (a SoxS homolog) is abolished by spacing changes of 1-2bp.

Reviewer #3:

Remarks to the Author:

In the current manuscript, the authors build on the findings of a previous manuscript to establish additional site requirements for effective CRISPRa-mediated gene activation. The study is motivated by the finding that application of CRISPRa towards endogenous genes using previous site requirements failed to produce significant activation. The authors go on to pose several hypotheses and design and perform systematic experiments to understand the cause of the limitations. Requirements that result from this systematic study provide important insights for successful implementation of CRISPRa. Overall, this manuscript is a well-written and logical presentation of a thorough and systematic study. Importantly (and somewhat refreshingly), the authors do not oversell the conclusions from their work. While they highlight the importance of periodicity within a narrow (40-bp) window upstream of the transcription start site, they also clearly report the failure to predictably activate gene expression with certain targets that have been guided by their studies. As such, the work should be considered an important next step but not a definitive elucidation of the design principles for CRISPRa in bacteria.

The manuscript would benefit from addressing the following points:

1. Line 25 – The meaning of the word “flexibly” is not clear in this context. Please be more specific.
2. Line 29 – Consider writing “To programmably down-regulate target genes...” to make it clear gRNAs and scRNA are used for CRISPRi and CRISPRa, respectively.
3. Line 146 – The wording here could be more specific. We suggest “Both systems effectively activate expression from synthetic and heterologous promoters,...”.
4. Figure 2 – The captions appear to be incorrect for panel A. Positions A1 and A2 are outside of the -60 to -100 bp window, while A3 and A4 are inside the window.
5. Figure 3B – What is the significance of the bold face type vs lowercase vs colored single nucleotides?
6. Figure 3D – The plotted data or bar arrangement could be modified to match the description of the experiment in the text (Lines 164-166). We suggest either grouping bars corresponding to samples with the same TetR/aTc conditions together or reporting fold activation based on specific and non-specific scRNA levels for each TetR/aTc treatment. Also in this panel, the dots representing individual replicates are not clearly visible.
7. Figure 6B/Supplemental Figure 9 – The disagreement in the trends between these two figures raises concern about the GFP assay in Figure 6B. If available, please provide a brief explanation for the possible cause of this discrepancy.
8. Supplementary Figure 1 – There are no black dots indicating replicates as described in the figure caption.

Response to reviewer comments

NCOMMS-19-1125140

We thank the reviewers for their detailed comments. We have addressed each point as described below. Our responses are in blue. Changes to the manuscript have been highlighted with “Track Changes” in the document.

Reviewer #1 (Remarks to the Author):

In their manuscript, “Effective CRISPRa-Mediated Control of Gene Expression in Bacteria Must Overcome Strict Target Site Requirements”, Fontana and collaborators present new insights into the use of a dCas9 tethered SoxS effector as a bacterial CRISPRa system. Despite some interesting observations, their findings are fundamentally negative results that call into question the existence of any reasonable use cases for CRISPRa. Additionally, the authors’ results about the extreme spacing dependence of CRISPRa should have been expected, and frankly should have been addressed in their initial (2018) paper. I detail these points further below.

Our manuscript presents both negative and positive results. We feel that it is critically important to present the negative results that clearly and systematically define the rules for bacterial CRISPRa. We note that Reviewer #3 agrees with this perspective. Further, these results open new and fundamental research questions about bacterial gene activation.

While we recognize that our results highlight challenges for predictable gene activation in bacteria, we have identified general rules and used these rules to successfully activate several endogenous genes. We are optimistic that further research will be able to provide predictive rules to enable routine CRISPRa in bacteria.

1. The authors’ further characterization of CRISPRa undermines many use cases for CRISPRa. The clearest use case for CRISPRa over another method of over-expression (such as cloning onto a plasmid, or cloning in a different promoter) is for screening.

We believe that CRISPRa screens will be possible and are currently performing pilot studies, although we suggest that these studies are well beyond the scope of this work. Further, there are many additional use cases for CRISPRa systems in bacteria. For example, there is an extensive literature on using CRISPRa/i tools to regulate heterologous gene circuits, including microbial papers published recently in Nat Comm (Liu et al., 2019 doi:10.1038/s41467-019-11479-0; Gander et al. 2017 doi:10.1038/ncomms15459). Conceptually similar work has been described in mammalian cells, underscoring the value of these types of tools for general applications in biological circuit design (Kiani et al., 2014 doi:10.1038/nmeth.2969; Nissim et al., 2014 10.1016/j.molcel.2014.04.022).

a. The need to target promoters diminishes the value of CRISPRa. To be able to perform useful CRISPRa experiment, one must know the promoters of the genes that one wants to target. Even well studied organisms like E. coli have incompletely characterized promoters. Many genes have

multiple (sometimes overlapping) promoters. Targeting the wrong promoter could readily lead to repression (by interfering with the active promoter) instead of activation.

We acknowledge that there are remaining challenges associated with CRISPRa in bacteria, and we suggest that systematic characterizations of the sort described here are critically important for surmounting the issues raised by the reviewer and advancing the field. Although there are unique challenges in bacteria, many of the points raised by the reviewer above apply to CRISPRa in any organism. There is still substantial value and broad interest in developing these tools for programmable gene regulation.

b. The authors show that the sequence between -60 and -35 makes a huge difference in CRISPRa activity but provide no information (other than speculation) on how this could happen. So even if all of the above criteria could be met, one would still have no way of knowing whether the system was actually working.

It is possible to determine whether the system is working using reporter gene assays with endogenous promoters, or direct assays like RT-qPCR on endogenous target genes. Nevertheless, the reviewer is correct that we lack a clear explanation for why sequence variation between -60 and -35 has such a large effect on activity, and it is not clear how to predict whether a given sequence is problematic. To assess this question, we sequenced a sampling of 29 sequence variants spanning a range of CRISPRa activity levels. We had speculated that transcription factor binding sites might appear, and we found that many of the variable intervening sequences had motifs that were within one base of a known consensus binding site. There was no obvious correlation between CRISPRa activity level and the presence of these sites, but it is not known which of these motifs actually bind endogenous transcription factors. Sequence logo plots reveal that ineffective sequences tend to be more AT-rich, although again we lack a clear explanation for why AT-rich intervening sequences should effect CRISPRa. We have added these points to the manuscript (pg 7, Supplemental Figure 3, Supplemental Table 6).

c. Because of the strict target site requirements, only some genes have PAMs that could be successfully targeted. This is partially addressed by dxCas9. However, off-target effect of this variant in CRISPRa system should be evaluated.

We respectfully suggest that additional experiments to test the off-target activity of dxCas9 are not necessary to support the central conclusions of our manuscript. The original paper describing xCas9 reported that its specificity was actually improved compared to *S. py* Cas9 (Hu et al., 2018). Also, our major conclusion from the dxCas9 experiment is that access to additional target sites enables CRISPRa at genes that were previously inaccessible in bacteria. New Cas9 variants and orthologs are being introduced very frequently, and there are many options that can be explored in future work with varying PAM compatibility and off-target activity.

2. It should have been apparent that strict spacing would be required for SoxS function. Almost all Class I activators (those that don't overlap the -35) function by interacting with the C-terminal domain of the alpha subunit of RNAP (αCTD). An extensive body of literature has shown that

these activators must be on the same face of the DNA helix as RNAP to function. For example, in (doi:10.1093/nar/gku625), the ability of CRP to activate a promoter across different spacings is tested and shown to have DNA periodicity. Similarly, in (<https://doi.org/10.1046/j.1365-2958.1999.01599.x>), it is shown that activation by MarA (a SoxS homolog) is abolished by spacing changes of 1-2bp.

We acknowledge that there is precedent for the phase dependence of transcriptional activation in bacteria. We discussed this point and prominently cited three representative publications in our original manuscript, including the first reference suggested by the reviewer above (doi:10.1093/nar/gku625). We also showed that activation by wt SoxS at a SoxS-dependent promoter is abolished by spacing changes of 1-2 bp (Supplementary Figure 7), reproducing a result previously reported in the literature (Wood et al., 1999; also cited in our original manuscript). We have added five additional citations to the manuscript to support this point, including the second reference suggested by the reviewer above (doi: 10.1046/j.1365-2958.1999.01599.x). We have also moved this discussion earlier in the manuscript to page 10.

We respectfully disagree that the sharp phase dependence we observed was obvious. SoxS is tethered to the CRISPR complex by a flexible linker, and it is genuinely surprising that, despite multiple strategies described in the manuscript, we were unable to modify the CRISPRa complex and broaden the observed phase dependence.

Reviewer #3 (Remarks to the Author):

In the current manuscript, the authors build on the findings of a previous manuscript to establish additional site requirements for effective CRISPRa-mediated gene activation. The study is motivated by the finding that application of CRISPRa towards endogenous genes using previous site requirements failed to produce significant activation. The authors go on to pose several hypotheses and design and perform systematic experiments to understand the cause of the limitations. Requirements that result from this systematic study provide important insights for successful implementation of CRISPRa. Overall, this manuscript is a well-written and logical presentation of a thorough and systematic study. Importantly (and somewhat refreshingly), the authors do not oversell the conclusions from their work. While they highlight the importance of periodicity within a narrow (40-bp) window upstream of the transcription start site, they also clearly report the failure to predictably activate gene expression with certain targets that have been guided by their studies. As such, the work should be considered an important next step but not a definitive elucidation of the design principles for CRISPRa in bacteria.

We thank the reviewer for their constructive comments and we have addressed their specific points as indicated below.

The manuscript would benefit from addressing the following points:

1. Line 25 – The meaning of the word “flexibly” is not clear in this context. Please be more specific.

We removed the word flexibly to avoid any confusion.

2. Line 29 – Consider writing “To programmably down-regulate target genes...” to make it clear gRNAs and scRNA are used for CRISPRi and CRISPRa, respectively.

3. Line 146 – The wording here could be more specific. We suggest “Both systems effectively activate expression from synthetic and heterologous promoters,...”.

We thank the reviewer for the comments, we have implemented the suggested edits.

4. Figure 2 – The captions appear to be incorrect for panel A. Positions A1 and A2 are outside of the -60 to -100 bp window, while A3 and A4 are inside the window.

We have corrected the error.

5. Figure 3B – What is the significance of the bold face type vs lowercase vs colored single nucleotides?

The bold letters represent the -35 and -10 regions of the minimal promoters. We have added a line to clarify this point in the figure legend. The red colored letter is the annotated TSS, we have updated the figure to show only the sequences up to the TSS.

6. Figure 3D – The plotted data or bar arrangement could be modified to match the description of the experiment in the text (Lines 164-166). We suggest either grouping bars corresponding to samples with the same TetR/aTc conditions together or reporting fold activation based on specific and non-specific scRNA levels for each TetR/aTc treatment. Also in this panel, the dots representing individual replicates are not clearly visible.

We thank the reviewer for the suggestion. We have updated the figure.

7. Figure 6B/Supplemental Figure 9 – The disagreement in the trends between these two figures raises concern about the GFP assay in Figure 6B. If available, please provide a brief explanation for the possible cause of this discrepancy.

We repeated this assay to investigate the discrepancy. Using flow cytometry, we found that some samples had bimodal fluorescence distributions, possibly indicative of contaminating mixed populations. We reconstructed the strains and repeated all samples using a flow cytometry assay, and obtained updated fluorescence data that agrees with the RT-qPCR data shown in Supplemental Figure 9. We have updated this figure to show the data collected by flow cytometry. We thank the reviewer for highlighting this discrepancy.

8. Supplementary Figure 1 – There are no black dots indicating replicates as described in the figure caption.

We thank the reviewer for the comment. There was a typo in the figure legend. This experiment was performed with one biological replicate as a technical triplicate. We updated the figure caption.

REVIEWERS' COMMENTS:

Reviewer #3 (Remarks to the Author):

The authors have addressed all of this reviewer's comments, save one. In lines 329-330, the authors note that the "better performing" sites for yajG and poxB were evaluated by qRT-PCR. Figure 6B shows that site Y2 had higher activation for yajG but Y1 is profiled in Supplementary Figure 11. I think this is the result of the repeat of experiments (in response to comment #7), which confirmed activated targets but resulted in a change in relative impact. This discrepancy should be addressed.

Response to final reviewer comments

NCOMMS-19-1125140

We have addressed the final reviewer comments as described below. Our responses are in blue.

Reviewer #3 (Remarks to the Author):

The authors have addressed all of this reviewer's comments, save one. In lines 329-330, the authors note that the "better performing" sites for yajG and poxB were evaluated by qRT-PCR. Figure 6B shows that site Y2 had higher activation for yajG but Y1 is profiled in Supplementary Figure 11. I think this is the result of the repeat of experiments (in response to comment #7), which confirmed activated targets but resulted in a change in relative impact. This discrepancy should be addressed.

The reviewer is correct, this discrepancy is a result of repeating the experiment in Figure 6B and we neglected to update the corresponding text. We have changed the text as follows: "We validated these results by performing RT-qPCR on the endogenous yajG and poxB loci. Targeting CRISPRa to these genes resulted in increases in RNA levels (Supplementary Figure 11)."